# Tree-Species Classification and Individual-Tree-Biomass Model Construction Based on Hyperspectral and LiDAR Data

Yifan Qiao [1,2], Guang Zheng [1,*], Zihan Du [3], Xiao Ma [1,2], Jiarui Li [1,2] and L. Monika Moskal [4]

1    International Institute for Earth System Science, Nanjing University, Nanjing 210023, China
2    Jiangsu Provincial Key Laboratory of Geographic Information Science, Nanjing 210023, China
3    Software and Big Data Department, Handan Polytechnic College, Handan 056001, China
4    Remote Sensing and Geospatial Analysis Laboratory, Precision Forestry Cooperative,
     School of Environment and Forest Science, University of Washington, Seattle, WA 98195, USA
*    Correspondence: zhengguang@nju.edu.cn

**Abstract:** Accurate classification of tree species is essential for forest resource monitoring, management, and conservation. Based on the classification of tree species, the biomass model at the individual-tree scale of each tree species can be accurately estimated, which can improve the estimation efficiency of individual-tree biomass. In this study, we first extracted four categories of indicators: canopy height model, spectral features, vegetation indices, and texture features from airborne-laser-scanning (ALS) data and hyperspectral data. We used these features as inputs to the random forest algorithm and screened out the optimal variable combination for tree-species classification, with an overall accuracy of 84.4% (kappa coefficient = 0.794). Then, we used ALS data to perform tree segmentation in forest plots to extract tree height, crown size, crown projected area, and crown volume. According to multivariate nonlinear fitting, the parameters of the individual-tree structure were introduced into the constant allometric ratio (CAR) biomass model to establish the biomass models of three tree species: Douglas fir, Red alder, and Bigleaf maple. The results showed that the model-fitting effects were improved after introducing the crown parameters. In addition, we also found that better tree segmentation results led to more accurate structural parameters.

**Keywords:** airborne-laser-scanning data; hyperspectral image; tree-species classification; tree segmentation; individual-tree-biomass model

## 1. Introduction

Understanding the composition of forest tree species can provide valuable information for estimating the economic value of forests and studying forest ecosystems [1]. Correct identification of forest tree species was of great significance for rational planning and utilization, sustainable forest management, biodiversity monitoring, and ecological environment protection [2]. Meanwhile, the accurate identification of forest tree species also provided reliable help for the estimation of forest biomass. Accurate estimation of forest biomass played a crucial role in afforestation planning and design, forest resource monitoring, forest ecological-value assessment, and climate-change impacts [3,4]. Therefore, efficient and practical ways of classifying tree species and constructing a single-tree-level biomass model based on the structure information of each tree species were highly desirable.

Traditional field surveys consumed a lot of manpower and material resources, the cost of which was too high to be implemented in a large area [5]. At present, various remote-sensing data have been used for tree-species classification such as multispectral, hyperspectral, and light detection and ranging (LiDAR) data [6,7]. Hyperspectral remote-sensing technology was a breakthrough in the field of remote sensing, which realizes the acquisition of a continuous and large range of spectral feature information. Compared with multi-spectrum remote-sensing data, hyperspectral data have the characteristics of high spectral resolution and continuous bands, which greatly reduces the phenomenon of

"same object different spectrum" and "foreign body same spectrum", and greatly improves the ability of ground object recognition [8,9]. Airborne laser scanning (ALS) can extract high-precision forest-tree height, while also providing information on the forest canopy surface's horizontal distribution and vertical structure [10–13]. Alonzo et al. [5] extracted all spectra exceeding the normalized vegetation index threshold from hyperspectral data and crown structural metrics from ALS data for classification to map the distribution of 29 common tree species in Santa Barbara, California, USA. From this, it can be known that the classification of multiple tree species can be achieved only by using optical remote-sensing data to extract vegetation indices. Liu et al. [14,15] used the method of combining ALS and hyperspectral data to classify tree species, which had higher accuracy than only using hyperspectral data. This indicated that adding vertical structure information can effectively improve the accuracy of tree-species classification. Based on the accurate classification result of tree species, it was helpful to determine the tree species corresponding to a single tree in the region, so as to build the biomass model of different tree species. Therefore, the combination of hyperspectral data and ALS data can obtain better tree-species-classification results and improve the efficiency of forest-resources monitoring in remote-sensing forestry to a certain extent.

LiDAR data can be used to obtain tree structure information, such as the tree height, diameter at breast height (DBH), and crown diameter, so as to fit the appropriate biomass model. Kankare et al. [16] constructed a new biomass model based on tree height and canopy structure extracted from terrestrial laser-scanning data and individual-tree-component biomass, which showed that LiDAR data could improve accuracy for biomass model construction. It indicated that this method greatly reduced the fieldwork of tree biomass measurement. After the establishment of the model, the biomass of standing forest can be estimated. Usoltsev et al. [17] constructed the new allometric equation with virtual variables to estimate birch's individual-tree and stand biomass. When estimating the biomass model of individual-tree levels, the commonly used parameters were height and DBH. However, it was difficult to guarantee the estimation accuracy with only two variables in the biomass model [18]. Liu et al. [19] added two parameters of crown projected area and crown volume to build the biomass model on the basis of the above two parameters. The results showed that the accuracy of models was improved. For the research of biomass models, scholars have proposed many biomass models which were summarized into three types: linear model, nonlinear model, and polynomial model. Nonlinear models are the most widely used, of which the relative growth model, constant allometric ratio (CAR) model, and variable allometric ratio (VAR) model were the most representative. By resampling and comparing the two types of model, the CAR model had not only a stable parameter-estimation value but also strong estimation ability [20]. Feng et al. [21] believed that the crown factor must be taken into account when building a biomass model, because the crown is an important part of the whole tree. Thus, they introduced crown volume and surface area into the biomass model. The results showed that the introduction of crown parameters into the biomass model could obtain higher accuracy compared with the traditional CAR model. Therefore, to build a high-precision biomass model, it was necessary to consider the crown factor as a parameter and select the CAR model for fitting the model.

According to the above, most studies [8–21] have focused only on tree-species classification or biomass model estimations by combining optical remote-sensing data and LiDAR data, while they have rarely paid attention to linking the two. In this study, we first combined active and passive remote-sensing data to realize tree-species classification. We then segmented individual trees according to the distribution range of trees and extracted individual-tree structural parameters. Finally, biomass models of each tree species were estimated at the individual-tree level. Therefore, the specific goals of this study were that we first investigated the optimal variable combination of tree-species classification based on ALS and hyperspectral data. Then, we developed a practical parameter-extraction algo-

rithm for individual-tree structural parameters based on the results of tree segmentation. Finally, the estimation results of biomass models of three species were discussed.

## 2. Materials and Methods

### 2.1. Study Area

We selected a natural forest area in Panther Creek (PC, 45.28°N, 123.37°W) in Oregon, USA (Figure 1a). PC is located on the northwestern coast of the USA, with an altitude range of 90.44 m to 703.14 m. The climate is a temperate maritime climate, with an average annual temperature of 15 °C. The total area of the study area is about 2580 ha. The types of vegetation in the study area are diverse, of which most of the vegetation is tall trees, and a few is understory vegetation such as shrubs and grasses. The study area contains 13 tree species, of which three tree species account for more than 88%, including Douglas fir (*Genus Pseudotsuga*), Red alder (*Alnus rubra*), and Bigleaf maple (*Acer macrophyllum*) [22].

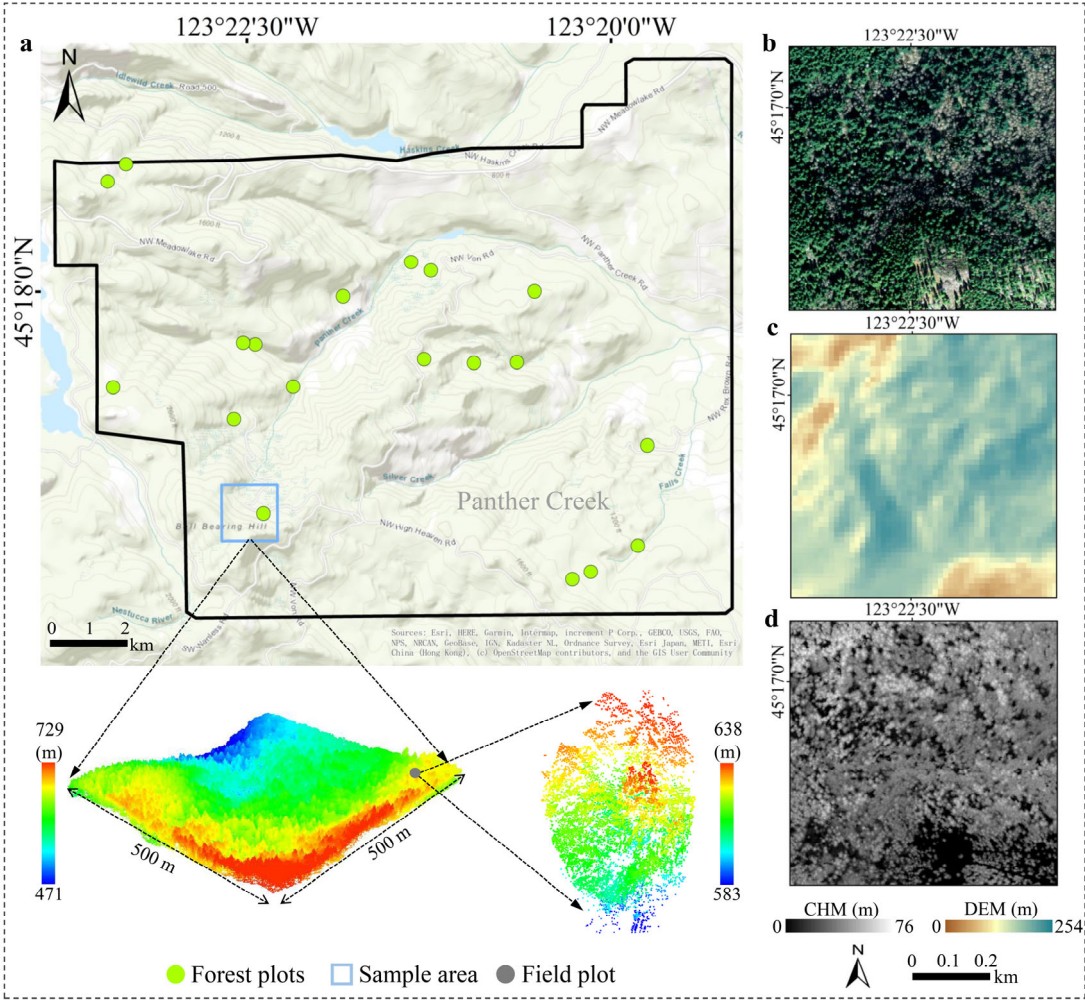

**Figure 1.** (**a**) The location of the study area was in Panther Creek, Oregon in northwest USA; (**a**) also had sample area's point clouds, (**b**) was the hyperspectral image, (**c**) was the digital elevation model (DEM), and (**d**) was the canopy height model (CHM) of the sample area.

### 2.2. Datasets

2.2.1. Airborne-Laser-Scanning Data

The ALS data were acquired by a Leica ALS60 sensor (Leica Geosystems AG, Heerbrugg, Switzerland) mounted in a small multipurpose aircraft to undertake the flight mission on July 15, 2010. The data-acquisition day was clear and not very cloudy. The projection coordinate system of the data was set as UTM10N with the horizontal and vertical

datums as NAD83 and NAVD88 (unit: meters), respectively. Each laser point contains the coordinates, height, intensity, classification, and echo-type information.

### 2.2.2. Remotely Sensed Optical Image

The hyperspectral remote-sensing image (Figure 1b) was acquired by a HyMap hyperspectral imager (HyVista Corporation, New South Wales, Australia) on 15 August 2010, including 125 bands with a spectral range from 400 nm to 2500 nm (Table 1).

**Table 1.** ALS and hyperspectral data-acquisition parameters.

| Category | Parameter | Name and Value |
|---|---|---|
| ALS data | Sensor | Leica ALS60 |
| | Field of view (°) | 28 |
| | Flight height (m) | 900 |
| | Pulse rate (kHz) | 105 |
| | Accuracy (m) | 0.03 |
| | Overlap | 100% (50% side-lap) |
| | Average point density (points/m$^2$) | 20 |
| Hyperspectral data | Sensor | HyMap |
| | Spectral range (nm) | 400~2500 |
| | Spectral resolution (nm) | 15~16 |
| | Spatial resolution (m) | 2.9 |
| | Field of view (°) | 61.3 |

Note: ALS: airborne laser scanning.

### 2.2.3. Field-Measured Data

In July 2009, we obtained some trees' actual data including tree species, tree number, DBH, height to live crown base (HTLB), coordinates, and tree-height information in PC (Table 2). We selected a total of 19 circular forest plots with a radius of 16 m in PC. The measured trees were located in these plots. The forest types include coniferous forest, broadleaf forest, and mixed coniferous and broadleaf forest. We selected a total of 90 (3 × 30) trees from three tree species in these forest plots to screen the optimal variables for tree-species classification.

**Table 2.** Forest plot characteristics in the PC study area.

| Plot | Tree Number | Forest Type | Mean DBH (cm) | Mean Height (m) | Mean HTLB (m) |
|---|---|---|---|---|---|
| PC-1 | 65 | C | 13.85 | 9.88 | 1.19 |
| PC-2 | 44 | M | 10.38 | 7.43 | 1.37 |
| PC-3 | 43 | M | 20.27 | 17.36 | 9.94 |
| PC-4 | 43 | C | 29.56 | 25.14 | 13.56 |
| PC-5 | 28 | M | 31.19 | 25.65 | 15.99 |
| PC-6 | 78 | M | 16.04 | 7.34 | 5.86 |
| PC-7 | 31 | M | 40.82 | 27.56 | 17.47 |
| PC-8 | 38 | M | 31.52 | 21.13 | 11.94 |
| PC-9 | 45 | M | 29.95 | 24.41 | 15.60 |
| PC-10 | 46 | M | 39.42 | 26.93 | 16.39 |
| PC-11 | 50 | C | 27.67 | 21.61 | 14.60 |
| PC-12 | 73 | C | 6.66 | 5.23 | 0.02 |
| PC-13 | 22 | B | 44.00 | 23.50 | 13.97 |
| PC-14 | 43 | M | 24.16 | 20.96 | 12.28 |
| PC-15 | 60 | B | 18.01 | 17.80 | 10.26 |
| PC-16 | 49 | M | 21.88 | 13.60 | 7.79 |
| PC-17 | 68 | M | 26.84 | 25.20 | 19.06 |
| PC-18 | 73 | M | 18.99 | 17.26 | 9.63 |
| PC-19 | 121 | M | 15.65 | 14.82 | 8.05 |
| Mean | 54 | - | 24.57 | 18.57 | 10.79 |

Note: C: conifer; B: broadleaf; M: mixed; DBH: diameter at breast height; HTLB: height to live crown base.

### 2.3. Overall Work

The overall flowchart of the study is shown in Figure 2. We first preprocessed the raw ALS data to obtain the canopy height model (CHM). In addition, the spectral features (SF), vegetation indices (VI), and texture features (TF) were extracted from the high-spatial-resolution hyperspectral image. The extracted four types of indicator were trained by the random forest algorithm to obtain classification results and the evaluation of results. Moreover, we mapped the distribution of tree species in the whole study area according to the classification results of sample plots. After that, tree segmentation was carried out to determine tree species corresponding to individual trees. Finally, we extracted the structural parameters of various types of tree and built biomass models based on individual-tree levels.

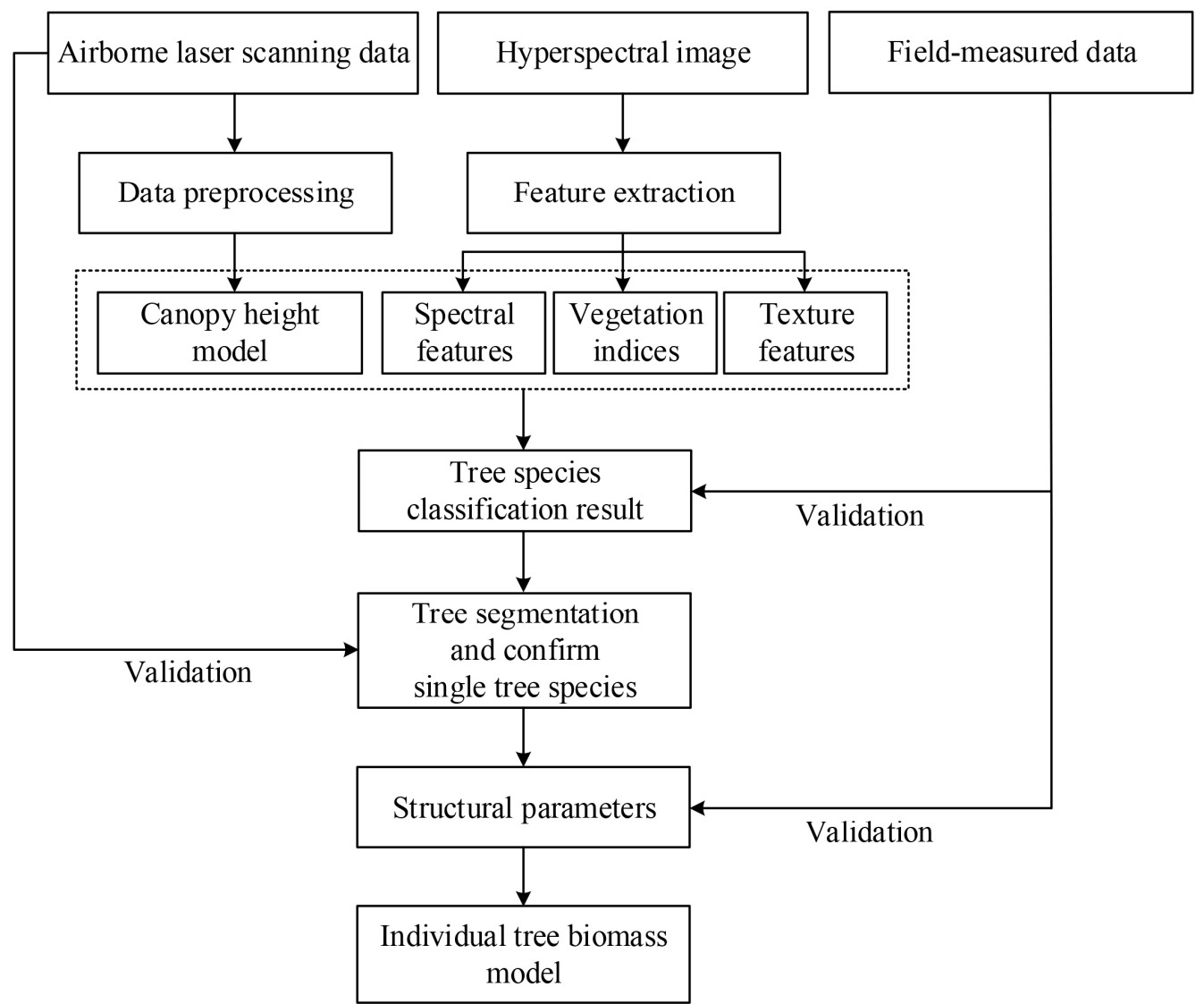

**Figure 2.** The flowchart of tree-species classification and biomass model construction based on airborne-laser-scanning data and hyperspectral image.

### 2.4. Data Preprocessing

The raw ALS data have been divided into ground and non-ground points by the data provider. We extracted a DSM by extracting the first echo of the non-ground point in Cloud Compare software (EDF, R&D, Paris, France), and its resolution was 0.5 m. Then, we used the inverse-distance-weighted (IDW) algorithm to generate a DTM with 0.5 m resolution from the last echo point of the ground point. The CHM was generated by the grid difference

operation between DSM and DTM in ArcGIS 10.6 software (Environment Systems Research Institute, Redlands, CA, USA). CHMs are able to show the spatial distribution of tree height and canopy. Finally, to eliminate the effect of terrain, we obtained the normalized non-ground point cloud by subtracting DTM from the z-coordinate of each non-ground point [23,24]. In addition, we also resampled the hyperspectral image to 0.5 m resolution using the nearest-neighbor method. We took a 500 × 500 m area as a sample area, and the processing results of LiDAR data are shown in Figure 1c,d.

### 2.5. Tree-Species Classification

### 2.5.1. Feature Extraction

Before the extraction of various features, we distinguished vegetation and non-vegetation in the study area. In this study, multiple vegetation samples and non-vegetation samples were extracted. By comparing their NDVI values, the threshold was determined as 0.45; that is, the pixels with NDVI $\geq$ 0.45 were vegetation, otherwise they were non-vegetation. Then, vegetation areas were extracted using ArcGIS10.6 software through the mask.

The data provider has performed systematic radiometric calibration and atmospheric correction on HyMap. Therefore, further parameter extraction can be carried out directly on the image. Different tree species can be distinguished according to their spectral characteristics. We extracted the reflectance values of sample points on each band as their SF in the ENVI 5.3 software (Exelis Visual Information Solutions, Boulder, CO, USA). Additionally, four vegetation indices (VI) were considered for the random forest algorithm, including the normalized difference vegetation index (NDVI1) using a near-infrared short band (NIR1) [25], NDVI2 using a near-infrared long band (NIR2) [26], green normalized difference vegetation index (GNDVI) [27], and enhanced vegetation index (EVI) [28]. The details of the VI are shown in Table 3.

**Table 3.** List of vegetation indices and texture features.

| Parameter Category | Index | Formula | Name and Description |
|---|---|---|---|
| Vegetation indices | NDVI1 | $(\text{NIR1} - \text{RED})/(\text{NIR1} + \text{RED})$ | Normalized Difference Vegetation Index-NIR1. |
| | NDVI2 | $(\text{NIR2} - \text{RED})/(\text{NIR2} + \text{RED})$ | Normalized Difference Vegetation Index-NIR2. |
| | GNDVI | $(\text{NIR1} - \text{GREEN})/(\text{NIR1} + \text{GREEN})$ | Green Normalized Difference Vegetation Index. |
| | EVI | $2.5 \times (\text{NIR1} - \text{RED})/(\text{NIR1} + 6 \times \text{RED} - 7.5 \times \text{BLUE} + 1)$ | Enhanced Vegetation Index. |
| Texture features | Energy | $\sum\limits_{a,b} P_{\phi,d}^2(a,b)$ | A measure of the stability of grayscale changes. |
| | Entropy | $\sum\limits_{a,b} P_{\phi,d}(a,b) log_2 P_{\phi,d}(a,b)$ | Randomness measures the amount of information contained in an image. |
| | Contrast | $\sum\limits_{a,b} \lvert a - b \rvert^k P_{\phi,d}^\lambda(a,b)$ | Grooves that reflect the clarity and texture of the image. |
| | IDM | $\sum\limits_{a,b;a \neq b} \dfrac{P_{\phi,d}^\lambda(a,b)}{\lvert a-b \rvert^k}$ | Reflect the homogeneity of the image texture and measure the local changes in the image texture. |

Note: NIR1: near-infrared short band (924 nm); NIR2: near-infrared long band (1700 nm); RED: the red band (708 nm); GREEN: the green band (546 nm); BLUE: the blue band (472 nm); a: the row number; b: the column number; $P_{\phi,d}$: the normalized value in the cell.

Table 3 shows four texture features including contrast, energy, entropy, and inverse differential moment (IDM) extracted from the hyperspectral image. These parameters can reflect the difference in different tree species' canopy contours [29]. This study obtained texture features from the Gray-Level Co-occurrence Matrix (GLCM). GLCM indicates the frequency at which different combinations of gray levels of two pixels at a fixed relative position occur in an image object [30]. The grayscale co-occurrence matrix is able to provide changes in the gray value of the image, including direction, interval, and magnitude of change. In practical applications, the extraction effect of texture features had a very good relationship with the image resolution. At the same time, the extraction of texture features needed to set different distance directions. Generally, the distance was set unchanged, and

the texture information in different directions was calculated or the texture information in multiple directions was calculated. The average value was taken as the gray value [31].

### 2.5.2. Optimal Variable Selection

In this paper, we extracted four types of feature, CHM, SF, VI, and TF, which increased the dimension of classified data. SF variables may be highly correlated or redundant due to the large number of bands that may increase the complexity of calculation. This made the classifier unable to play its role and even caused accuracy degradation [32,33]. The main feature of the random forest (RF) algorithm is that it does not produce an overfitting phenomenon when processing high-dimensional data. The RF algorithm is a classifier with multiple decision trees, which has a high classification accuracy. For this reason, we chose the RF algorithm to screen variables.

We used the RF function in the R language to implement tree-species classification. Through experiments, the number of features of each node used the square root of the total number of features. The number of binary tree variables and decision trees were the default values. We first used four types of feature indicator as categorical variables separately to classify tree species, and then sorted all feature variables by the importance indicator—the Gini index. The low-importance-ranked feature variables were removed.

### 2.6. Individual-Tree Biomass Model

#### 2.6.1. Tree Segmentation

We obtained the DTM from the data-preprocessing section and then used FUSION software to normalize the forest heights. Tree discrete point clouds were divided into individual tree crowns using the crown-segmentation algorithm proposed by Li et al. [34]. Therefore, we could obtain structural information about each tree, such as coordinates and tree height. The tree-crown-segmentation algorithm was a top-down region-growing method based on raw 3-D point cloud data. Minimum distance threshold (DT) and search sphere radius (SSR) for finding local maxima are two key parameters. In this study, we determined the DT and SSR as 2 m for plots [35].

In order to verify the results of tree segmentation, the visual tree crown with the location of the measured tree as a center point was obtained through visual interpretation. The x and y coordinates of the highest points of each segmentation tree were taken as the center point to draw the segmentation tree crown.

#### 2.6.2. Crown Parameter Extraction

According to the results of tree segmentation, the crown diameters in east–west and north–south directions were extracted. We calculated the average value as the individual tree-crown size. The tree-crown projection plane was the set of all points within a certain range, and the Graham two-dimensional convex hull was used to connect the most edge points of the point set to construct a convex polygon [36]. We estimated the projected area of the tree crown based on the area of the convex polygon (Figure 3). The calculation Formula (1) was as follows.

$$S = \frac{1}{2} \sum_{i=2}^{n} [X_i(Y_{i+1} - Y_i) - Y_i(X_{i+1} - X_i)] \tag{1}$$

where $S$ represented the area, m$^2$; $i$ represented the point number, $i = 2, 3, \ldots, n$.

The calculation method of tree-crown volume was based on integration [37]. The calculation was as follows in Formulaes (2)–(4). First, we divided the tree crown into several parts by layers, and a two-dimensional convex hull algorithm was implemented for each layer to calculate the cross-sectional area. The height between layers and cross-sectional area were used to calculate the volume of each part (Figure 3). Finally, all slice volumes were accumulated and summed to obtain the crown volume. The number of layers was determined according to the specific conditions of the crown of different tree species.

$$V_{Tc} = \frac{1}{3}\left(S_j + \sqrt{S_j S_{j+1}} + S_{j+1}\right) H_j \tag{2}$$

$$V_{Cc} = \frac{1}{3} S_j H_j \tag{3}$$

$$V = \sum_{c=1}^{n}(V_{Tc} + V_{Cc}) \tag{4}$$

where $j$ represented the section number, $j = 1, \ldots, n$; $S_j$, $S_{j+1}$ represented cross sectional area, m$^2$; $H_j$ represented height between two adjacent layers after segmentation; $V_{Tc}$ was the circular truncated cone; $V_{Cc}$ was the circular cone.

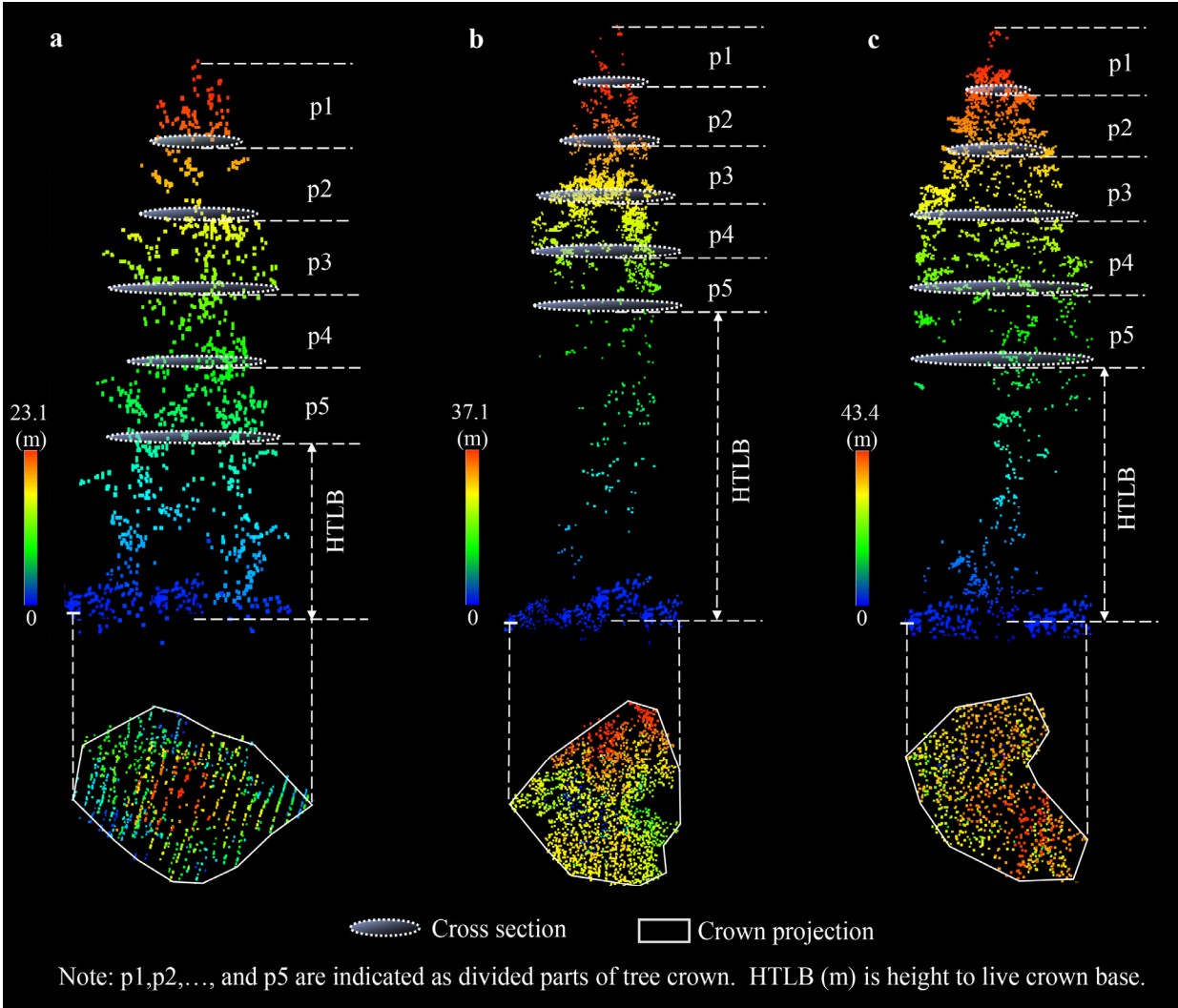

**Figure 3.** Individual tree's point cloud and crown projection convex hull of target trees in forest plots. Their crown stratification and cross-section are shown. (**a**) Douglas fir, (**b**) Red alder, and (**c**) Bigleaf maple.

### 2.6.3. Multivariate Nonlinear Fitting

The CAR model was stable and had good estimation accuracy [20,21]. Therefore, we chose it to study and construct biomass models of different tree species at the level of individual trees. The allometric equations for Douglas fir [38], Red alder [39], and Bigleaf maple [40] are shown in Table 4. In this study, based on the CAR model, we added tree height, crown size, crown projected area, and crown volume to construct biomass models. Since each crown parameter satisfied the relative growth relationship, the CAR model was

expressed as the multiplication of power functions of individual variables or combined variables. From the above, the biomass model including tree height, crown size, crown projected area, and crown volume was obtained as follows:

$$W = a_1 H^{a_2} C^{a_3} \tag{5}$$

$$W = a_1 H^{a_2} C^{a_3} S^{a_4} \tag{6}$$

$$W = a_1 H^{a_2} C^{a_3} V^{a_4} \tag{7}$$

$$W = a_1 H^{a_2} C^{a_3} S^{a_4} V^{a_5} \tag{8}$$

where $W$ was biomass, $H$ was tree height, $C$ was crown size, $S$ was crown projection area, $V$ was crown volume, and $a_1, a_2, a_3, a_4, a_5$ were model parameters.

**Table 4.** Allometric equations of the tree species.

| Tree Species | Allometric Equation |
|---|---|
| Douglas fir (*Genus Pseudotsuga*) | $W_D = 0.085982 \times DBH^{1.743391} \times H^{0.588628}$ |
| Red alder (*Alnus rubra*) | $W_R = 0.4026 \times DBH^{1.90612} \times H^{0.97674}$ |
| Bigleaf maple (*Acer macrophyllum Pursh*) | $W_{B-bark} = 2.338 \times DBH^{2.574}$ <br> $W_{B-stem} = 3.4148 \times DBH^{2.723}$ <br> $W_{B-leaf} = 0.4159 \times DBH^{2.5033}$ <br> $W_{B-live\ branch} = 2.6718 \times DBH^{2.43}$ <br> $W_{B-dead\ branch} = 4.7918 \times DBH^{1.092}$ |

Note: W: biomass of individual tree; DBH: diameter at breast height; H: tree height.

*2.7. Accuracy Assessment*

We used a confusion matrix to assess the classification accuracy of sample tree species. The producer's accuracy (PA), user's accuracy (UA), overall accuracy (OA), and kappa coefficient (Kappa) were used for accuracy assessment. The coefficient of determination ($R^2$) and the root-mean-square error (RMSE) were used to assess the fitting accuracy and predictive ability of models [41]. For the results after tree segmentation, we used three indicators: tree detection (TD), tree location RMSE, and crown radius RMSE to verify the segmentation accuracy of a single tree's point cloud. Tree detection was used to check the overall segmentation effect, as shown in Formula (11). We computed the standard deviations of the X- and Y- coordinates on the two-dimensional plane using Equation (12). By comparing the crown radius of visual interpretation with that of segmentation, we used Formula (13) to calculated the crown radius RMSE.

$$R^2 = \frac{\sum_{i=1}^{n} (x_i - \hat{x}_i)^2}{\sum_{i=1}^{n} (x_i - \overline{x_i})^2} \tag{9}$$

$$\text{RMSE} = \sqrt{\frac{\sum_{i=1}^{n} (x_i - \hat{x}_i)^2}{n}} \tag{10}$$

$$\text{TD} = \frac{N_{true}}{N_{seg-trees}} \times 100\% \tag{11}$$

$$\text{RMSE(X)} = \sqrt{\frac{\sum_{i=1}^{n} (X_i - X_i')^2}{n}} \tag{12}$$

$$\text{RMSE(r)} = \sqrt{\frac{\sum_{i=1}^{n} (r_i - r_i')^2}{n}} \tag{13}$$

where $\hat{x}_i$ was model prediction, $x_i$ was measured value, $\overline{x_i}$ was sample mean value, $N_{true}$ was the number of perfect segmentation trees, $N_{seg-trees}$ was the number of segmentation

trees, $X_i$ was the x coordinate of a field-measured tree's crown vertex, $X_i'$ was the x coordinate of a single tree's crown vertex obtained by segmentation, $r_i$ was a measured value by visual interpretation, $r_i'$ was the crown radius of a single tree obtained by segmentation.

## 3. Results

### 3.1. Results of Optimal Variable Selection

The spatial resolution of CHM used in this study was 0.5 m, and the resolution of the hyperspectral image was also 0.5 m. According to the position information of the measured tree, CHM can accurately express the tree height corresponding to each pixel on the hyperspectral image. The first column of Figure 4 shows the accuracy assessment results of Douglas fir, Red alder, and Bigleaf maple were classified with only CHM.

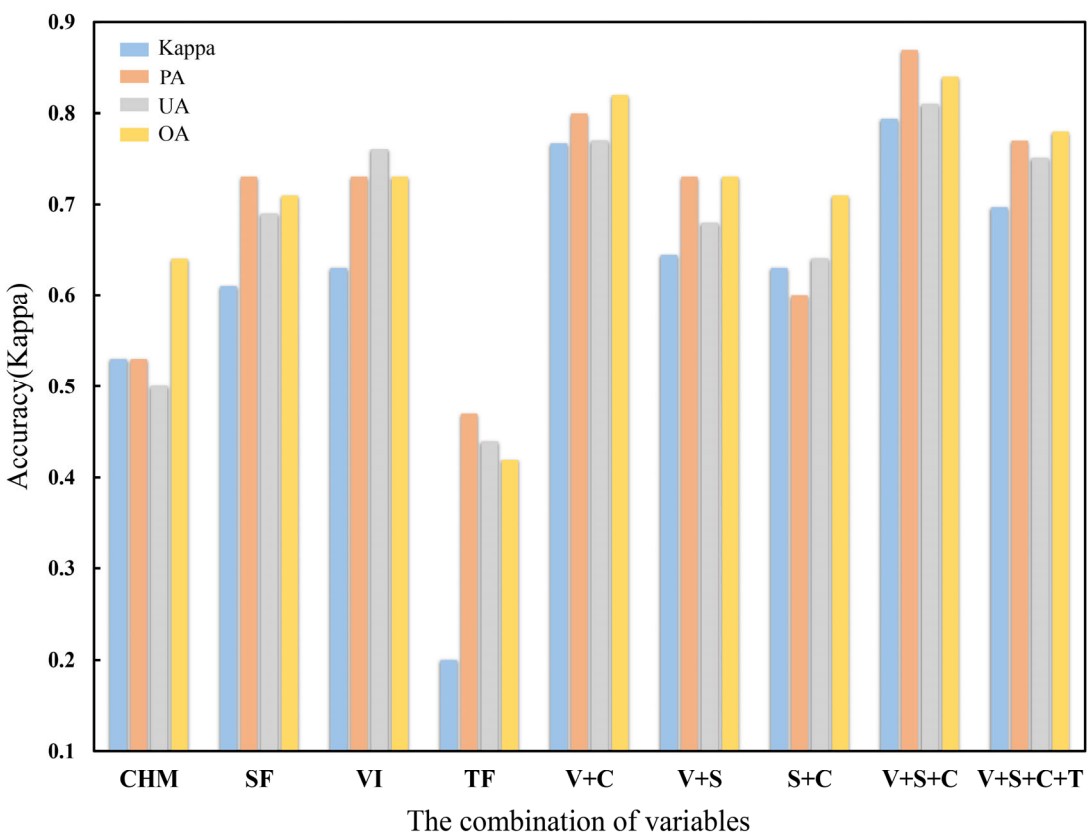

**Figure 4.** Accuracy assessment of tree-species classification with different combinations of variables. (Kappa: kappa coefficient; PA: producer's accuracy; UA: user's accuracy; OA: overall accuracy.)

The band characteristic curves of three tree species are shown in Figure 5a. From the difference and trend between the reflectivity curves, we selected the band values in the four ranges of band 1~15 (472~678 nm), band 22~52 (779~1206 nm), band 60~67 (1317~1462 nm), and band 100~112 (2062~2276 nm) to sort the variable's importance in the RF algorithm. According to the results of multiple tests and MDG, band 1 (472 nm), band 7 (561 nm), band 11~15 (620~679 nm), band 64~66 (1419~1448 nm), and band 102 (2098 nm) were initially selected. Moreover, in order to avoid the redundancy of variables with similar wavelength, we used Pearson correlation [42] to test band 11~15 and band 64~66. There was significant correlation ($p < 0.001$) between variables. According to the ranked importance by using spectral variables alone (Figure 6a), band 12 and band 66 were selected. Finally, band 1 (472 nm), band 7 (561 nm), band 12 (635 nm), band 66 (1448 nm), and band 102 (2098 nm) were selected to participate in classification. Using SF only as the classification variable, the accuracy is in the second column of Figure 4, which was more effective than using CHM alone.

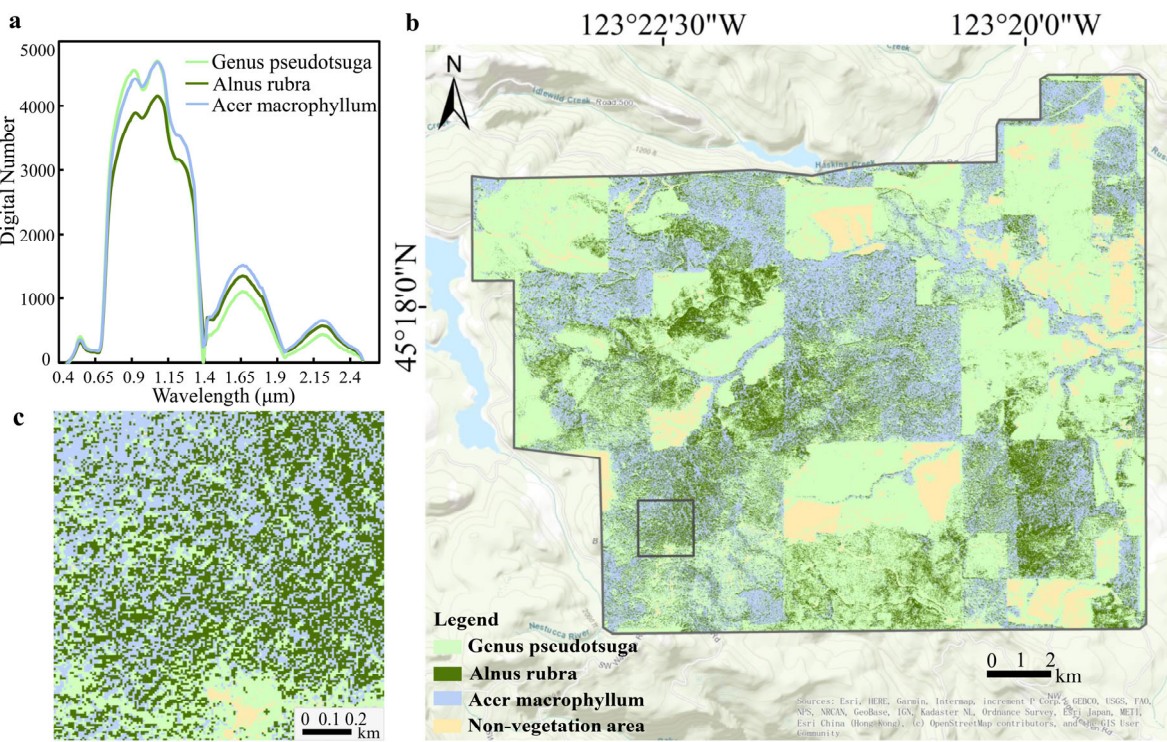

**Figure 5.** (**a**) The spectral reflectance of three tree species in PC. (**b**) The tree-species distribution map was derived from the optimal variable combination. (**c**) The classification result of tree species in sample plot.

In addition, NDVI1, NDVI2, GNDVI, and EVI were also screened according to order of importance. This indicated that NDVI2 was the most important, followed by GNDVI, EVI, and NDVI1. At the same time, four TF variables were selected: contrast, energy, entropy, and inverse difference matrix. Three tree species were also classified using only these four textural variables. The result of importance screening for the two types of variable separately is shown in Figure 6a. According to the accuracy result of variables of each type separately classified (Figure 4), VI was applied to classification, Kappa = 0.633 was the highest, followed by CHM, then SF and TF. We further screened the optimal variable combination by introducing variables step by step. Firstly, the importance of variables obtained by VI + CHM is shown in Figure 6b, the result of VI + SF is shown in Figure 6c, the result of CHM + SF is shown in Figure 6d, the result of VI + CHM + SF is shown in Figure 6e, and the result of VI + CHM + SF + TF is shown in Figure 6f. Next, we used the RF algorithm to classify tree species by combining the four categories of indicators from two data sources. The result is shown in Figure 4. The accuracy of Kappa, PA, OA, and UA obtained by the combination of VI + CHM + SF variables was the highest. Therefore, we concluded that VI + CHM + SF was the optimal combination of variables.

After obtaining the vegetation area, we further extracted CHM, VI, and SF corresponding to the location of each pixel according to the best combination of variables. The classification results of tree species in the whole study area were obtained (Figure 5b). It can be seen that roads, bare land, lakes, and a few buildings were well removed. By comparing the classification results with the measured samples, the overall classification accuracy of three tree species was 81%. Of these, the classification accuracy of Douglas fir was the highest at 93.3%, while that of Red alder was the lowest at 66.7%. The possible reason was that Red alder was mostly mixed with Bigleaf maple and the vegetation density of sample plots was very high. The leaves often overlapped, and the spectral information of Red alder was similar to that of Bigleaf maple. Therefore, it was not easy to distinguish Red alder from Bigleaf maple.

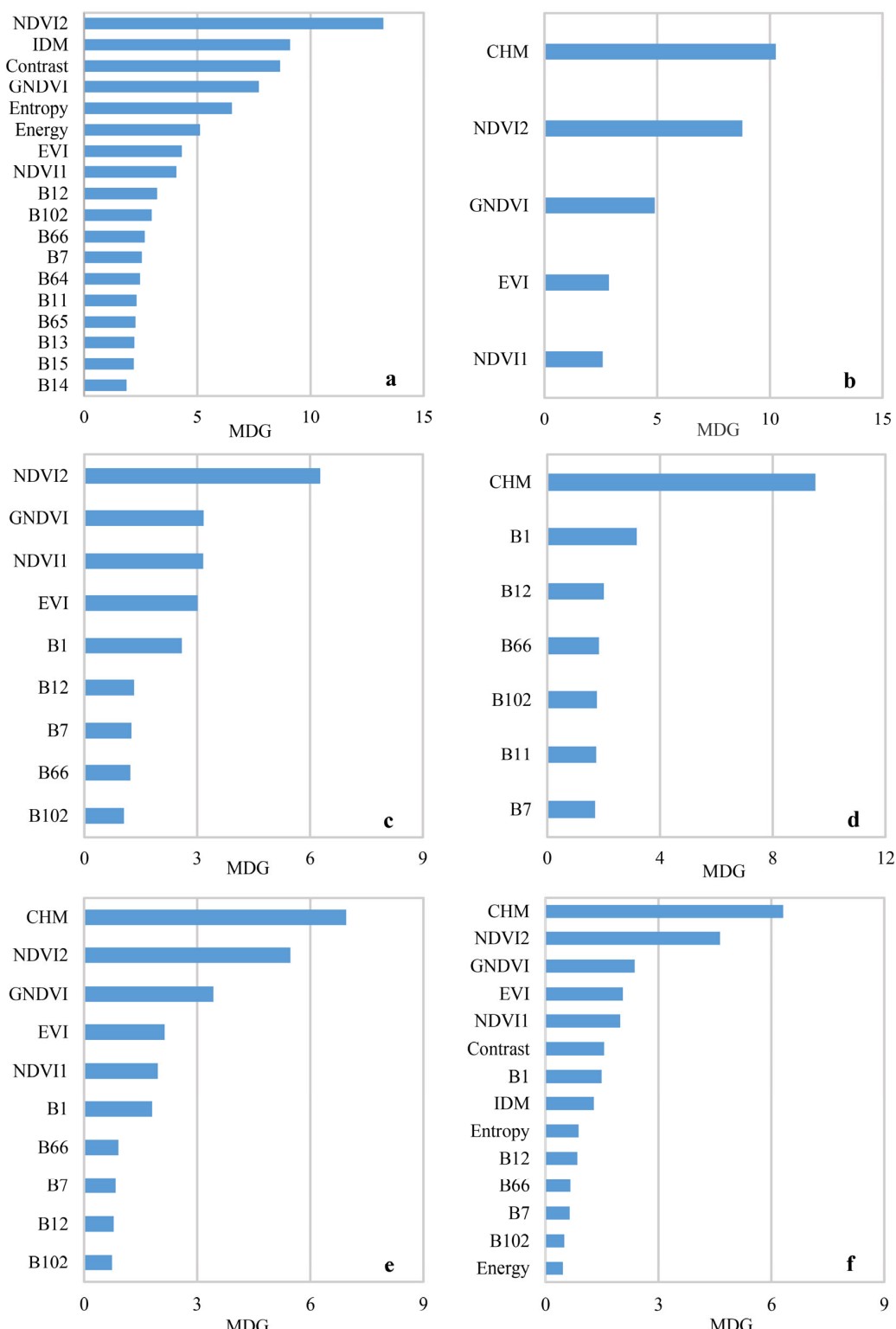

**Figure 6.** Variable importance ranking based on Mean Decrease Gini (MDG); (**a**) the variables' importance ranking results of using spectral features (SF), vegetation indices (VI), and texture features (TF), separately; (**b**) the canopy height model (CHM) combined with VI; (**c**) VI combined with SF; (**d**) CHM combined with SF; (**e**) CHM combined with VI and SF; and (**f**) the combination of four types of variables. (Note: NDVI1: normalized difference vegetation index using a near-infrared short band; NDVI2: normalized difference vegetation index using a near-infrared long band; GNDVI: green normalized difference vegetation index; EVI: enhanced vegetation index; IDM: Inverse Differential Moment; Bx: the reflectance value on band-x.)

### 3.2. Tree Segmentation and Validation

From the classification result map (Figure 5b), it can be seen that some regions of the single tree species can be seen. Then, we obtained tree segmentation results from the tree-species distributed area for seven types (Figure 7): medium-density conifer (MC), high-density conifer (HC), low-density mixed forest (LM), medium-density mixed forest (MM), high-density mixed forest (HM), low-density broadleaf (LB), and high-density broadleaf (HB). At the same time, we verified the segmentation results of an individual tree.

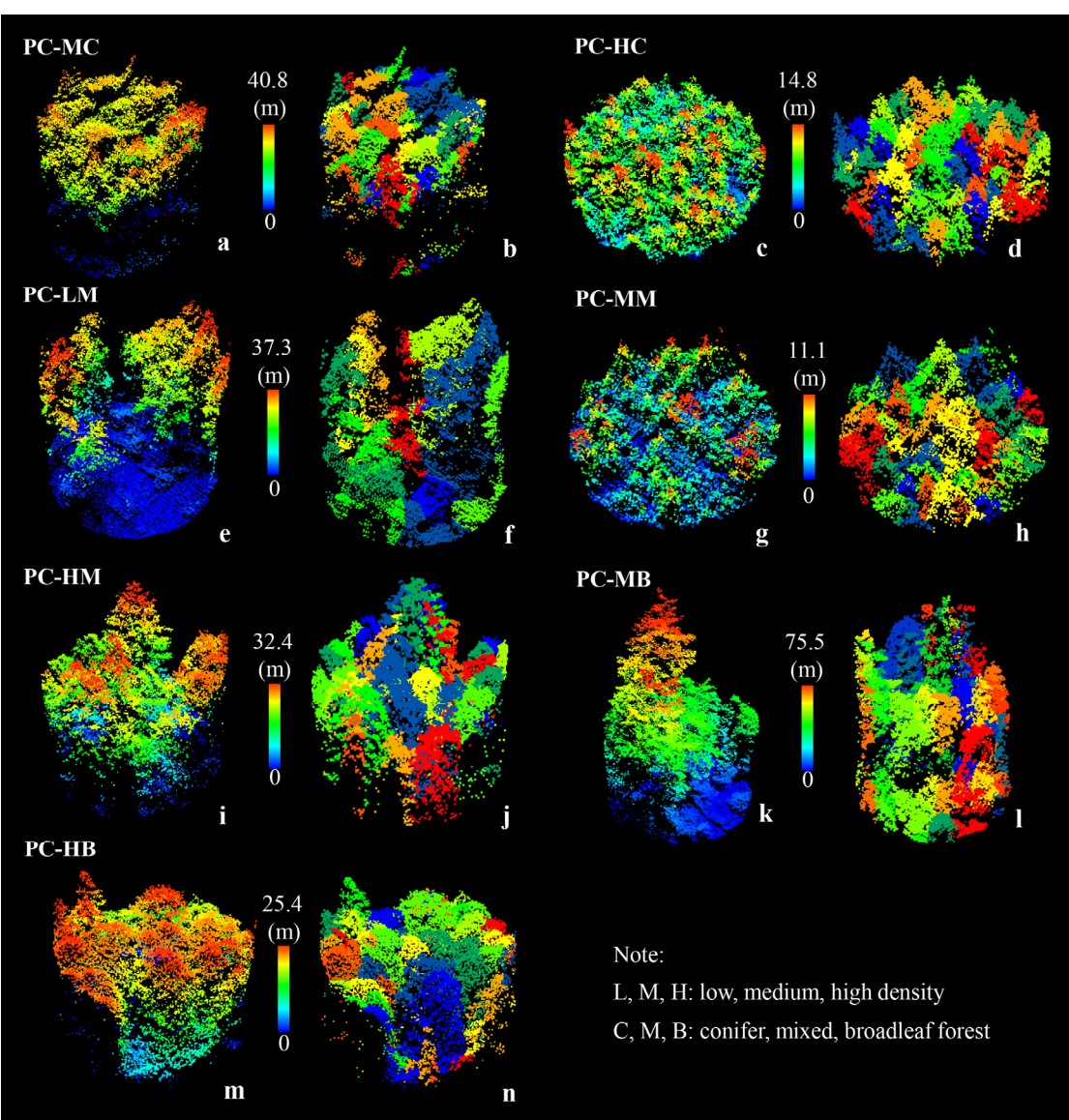

**Figure 7.** Height-normalized airborne-laser-scanning data (insets **a,c,e,g,i,k,m**) and corresponding tree-segmentation results (insets **b,d,f,h,j,l,n**) of seven types of forest plot. Different colors represented different trees in the tree-segmentation results.

As shown in Table 5, the tree location RMSE based on ALS increased with an increase in forest density in the sample plots with the same forest type. The RMSE of medium- and low-density coniferous forest, broadleaf forest, and mixed forest were between 0.72 m and 1.47 m, of which the RMSE of coniferous forest was the smallest. In addition, by comparing the canopy size of plots with different densities and forest types, it can be observed that the $R^2$ of plots with higher densities in the same forest type was smaller than that of plots with lower densities. The mixed forest of the same density was smaller than that of the

single-species stand. In this study, coniferous forests had well-separated crowns, while broadleaf forests and mixed forests often overlapped each other due to large crown sizes. Therefore, the RMSE of the tree location would be larger. Visually interpreted crown size based on CHM was used as validation data for individual-tree segmentation. According to the comparison results of tree species after tree segmentation, we selected a total of 90 (3 × 30) sample trees.

**Table 5.** Characteristics of the individual trees for forest plots.

| Plot ID | Density | Tree | Forest Type | Segmented Tree | Tree-Detection Rate | Tree Location RMSE (m) | Crown Radius RMSE (m) |
|---------|---------|------|-------------|----------------|---------------------|------------------------|------------------------|
| PC-MC | Medium | 181 | C | 167 | 90% | 0.72 | 1.88 |
| PC-HC | High | 50 | C | 46 | 87% | 1.52 | 2.3 |
| PC-LB | Low | 22 | B | 18 | 89% | 1.47 | 1.77 |
| PC-HB | High | 60 | B | 50 | 90% | 1.95 | 1.83 |
| PC-LM | Low | 72 | M | 66 | 92% | 1.32 | 1.65 |
| PC-MM | Medium | 78 | M | 56 | 89% | 1.40 | 1.2 |
| PC-HM | High | 557 | M | 468 | 84% | 2.65 | 1.73 |

Note: C: conifer; B: broadleaf; M: mixed; RMSE: root-mean-square error.

The regression relationship between the extracted tree height and the measured tree height of the three types of tree are shown in Figure 8. The results showed that there was a good correlation between the height of a single tree extracted by the proposed algorithm and the measured value. In summary, the correlation between the tree height from tree segmentation and the measured tree height of Douglas fir was the best, and Red alder had the least good results. Due to the large spacing between Douglas fir trees in the sample plot, there was not much overlap between trees. Therefore, good results would be obtained during tree segmentation, and the tree height would be closer to the true value. Most of the plots of Red alder were mixed forest with high vegetation density and overlapping trees. Thus, the tree height of individual trees was different from that of field-measured trees.

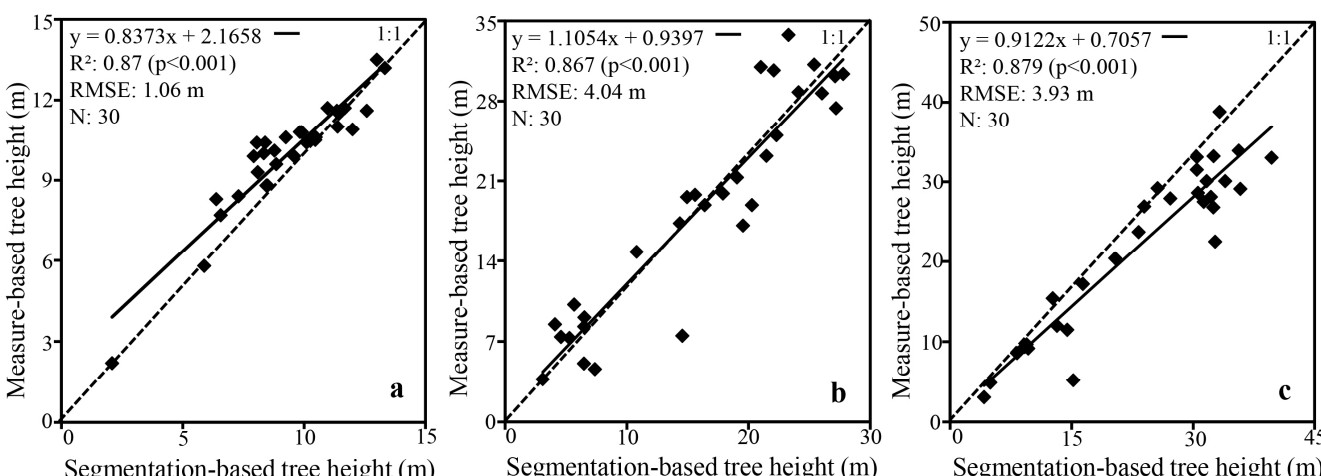

**Figure 8.** The regression relationship between the tree height extracted by tree segmentation and the measured tree height. (**a**) Douglas fir, (**b**) Red alder, and (**c**) Bigleaf maple.

### 3.3. Results of the Tree-Crown-Parameter Extraction

We obtained the crown projection area and crown-volume data statistics of Douglas fir, Red alder, and Bigleaf maple (Figure 9). On the whole, the crown projection area of Douglas fir was the smallest, with an average of 11.87 m$^2$ in the sample trees. Its crown size was relatively smaller than that of Red alder and Bigleaf maple belonging to the broad-leaved forest. The projection area of Red alder was 22.97 m$^2$, and the average projection area

was the largest of the three tree species. Moreover, the frequency of the projection area of Bigleaf maple was the largest at nearly 30 m². Thus, the crown size of Bigleaf maple was the largest, relatively. According to the results of crown volume, the mean value of Bigleaf maple was the largest. The distribution of the crown volume of Douglas fir in each interval was relatively uniform. This indicated that the sample tree size of this tree species was similar. Most of the Red alder's crown volumes were less than 60 m³, and the maximum value of 80.6 m³ corresponded to only one tree. There was no centralized area for the volume distribution of Bigleaf maple. There are five and eleven trees at the maximum and minimum volume values, respectively, and seven trees near the median of 200 m³.

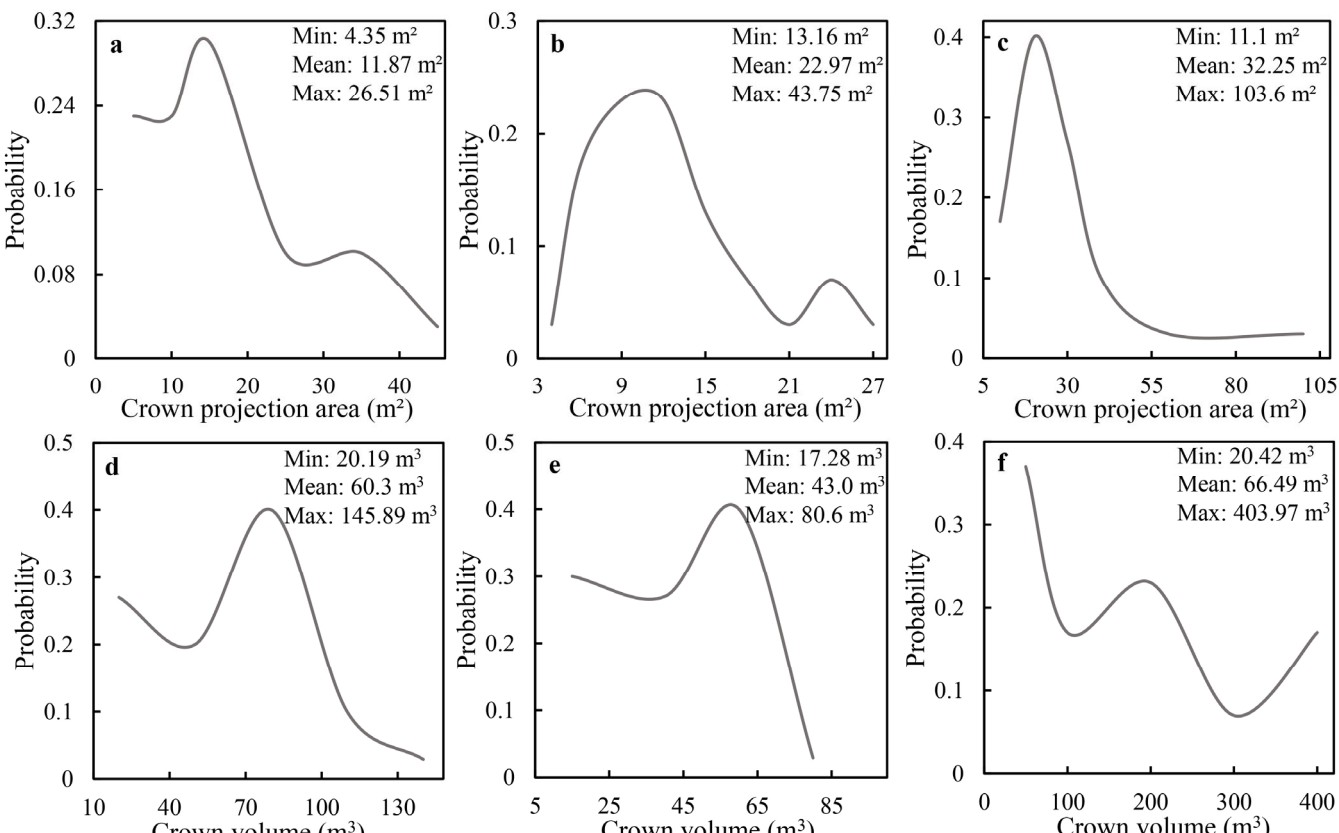

**Figure 9.** The crown projection area and volume of Douglas fir (**a**,**d**), Red alder (**b**,**e**), and Bigleaf maple (**c**,**f**).

### 3.4. Results of Individual-Tree-Biomass Model

The biomass was calculated based on the allometric-growth equation provided in Table 4. Combined with the single-tree parameters extracted from ALS data, and according to the biomass model Formulae (5)–(8), the nonlinear least-squares-regression method was used to analyze three tree species. According to the formulae, the tree height and crown size were first introduced into the model as independent variables, and the biomass as dependent variables. Then the projected area and volume of crown were introduced, respectively, and finally both were introduced at the same time.

Table 6 shows the fitting results of the biomass-estimation model, and the parameter estimation and accuracy evaluation. The goodness of fit of the three tree species increased with an increase in parameters. For Douglas Fir, when only tree height and crown size were involved in the fitting, the fitting accuracy was already good. However, when the crown projection area and crown volume were added, especially when the crown volume was added, $R^2$ increased by 0.031 and RMSE decreased by 1 kg·plant$^{-1}$. The same was true for Red alder. The optimal fitting results were obtained when we used Formula (8) in the fitting model, when $R^2$ increased by 0.082 and RMSE decreased by 0.95 kg·plant$^{-1}$. For

Bigleaf maple, $R^2$ significantly increased after the crown volume was added to the fitting model. $R^2$ increased by 0.294, and the RMSE decreased by 11.21 kg·plant$^{-1}$.

**Table 6.** Parameter estimation and goodness of fit of biomass models.

| Tree Species | Biomass Models | Parameter Estimation | | | | | Goodness of Fit | |
|---|---|---|---|---|---|---|---|---|
| | | a1 | a2 | a3 | a4 | a5 | $R^2$ | RMSE (kg·Plant$^{-1}$) |
| Douglas fir | 1 | 0.923 | 1.576 | 0.134 | - | - | 0.84 | 6.472 |
| | 2 | 0.78 | 1.607 | 0.105 | 0.055 | - | 0.843 | 6.083 |
| | 3 | 0.962 | 1.206 | 0.157 | 0.191 | - | 0.871 | 5.514 |
| | 4 | 0.888 | 1.232 | 0.141 | 0.185 | 0.027 | 0.871 | 5.477 |
| Red alder | 1 | 1.835 | 1.523 | 0.336 | - | - | 0.637 | 9.06 |
| | 2 | 2.52 | 1.412 | 0.355 | −0.065 | - | 0.654 | 9 |
| | 3 | 1.119 | 1.706 | 0.208 | 0.127 | - | 0.68 | 8.8 |
| | 4 | 0.028 | 1.559 | 0.165 | 0.182 | 1.241 | 0.709 | 8.11 |
| Bigleaf maple | 1 | 16.058 | −0.094 | 0.787 | - | - | 0.496 | 29.26 |
| | 2 | 6.453 | 0.002 | 0.336 | 0.422 | - | 0.601 | 27.71 |
| | 3 | 0.184 | 0.144 | 0.318 | 1.062 | - | 0.724 | 19.1 |
| | 4 | 0.008 | −0.383 | 0.075 | −0.038 | 2.184 | 0.790 | 18.05 |

Note: a1, a2, a3, a4, a5 were derived from Formulaes (5)–(8); RMSE: root-mean-square error.

## 4. Discussion

### 4.1. Feature Determination

Hollaus et al. [43] extracted single-tree-canopy height from LiDAR data, and the results showed that there was a good correlation between LiDAR tree height and field-measured tree height. Zhao [44] proposed an optimization method of CHM combined with canopy control, which made the accuracy of tree-height extraction greater than 90%. In this study, the study area was a large natural forest area. We extracted CHM from ALS data, which provided canopy distribution information. Accuracy results (Figure 4) were obtained when only using CHM for classification. Height information obtained with LiDAR data may not be enough to get good classification results, and large data samples combining multi-source remote-sensing data and deep learning will open up new possibilities for tree-species identification [45].

Hyperspectral remote sensing can provide rich spectral information to improve the discrimination ability of tree species [6]. HyMap can be used for tree-species classification and forest age division in coniferous forests in western Germany [46]. Li et al. [25] extracted multiple vegetation indices from high-resolution satellite imagery to classify tree species at the single-tree scale based on CNN. This study also considered spectral information for implementing tree-species classification. For the 125 bands of HyMap, we first divided three tree species into four band intervals with obvious distinctions, which were located in the green band, the red-band near-infrared short wave, and near-infrared long wave, respectively. Through the ranking of importance, multiple tests in the RF algorithm, and correlation tests, five bands, band 1 (472 nm), band 7 (561 nm), band 13 (650 nm), band 65 (1434 nm), and band 102 (2098 nm), were finally selected for classification. Through the above methods, the redundancy of variables could be reduced further. In the process of using each type of feature separately for classification, we first used all variables contained in the feature for tree-species classification, but the results were unsatisfactory. Then, according to the importance ranking, the variables with high importance have priority to participate in the classification, and the accuracy was significantly improved. Since low-importance variables did not help improve the classification accuracy, we strictly removed them. When we used the spectral information, VI, and the combination of the two, the classification accuracy results were better (Figure 4), which showed that the use of hyperspectral images was very beneficial for tree-species classification.

Pu et al. [30] calculated nine texture features including six GLCM and three GLDV based on band four of pan-sharpened multispectral images. In this study, we calculated

four GLCM textures based on the optical image. The lowest accuracy was obtained when only TF was used as classification metrics. When we used VI + SF + CHM + TF, the accuracy was improved, but it was not the optimal variable combination (Figure 4). We compared the crown shapes of the three tree species on the optical images. It was found that Douglas fir's shape and outline were more obvious in contrast to Red alder and Bigleaf maple, as it is a coniferous species. However, it was hard to distinguish in the areas with high forest density mixed between Red alder and Bigleaf maple. In short, TF as a single variable for tree-species classification may have low accuracy, but it can be used in combination with other types of variable to achieve better results. Therefore, when obtaining poor classification results by using only TF, other parameters should be considered to be introduced.

Wu [47] combined hyperspectral and LiDAR data sources to discuss five tree-species-classification schemes, each of which showed that applying multi-dimensional features can effectively improve tree-species-classification accuracy. Therefore, in this article, we extracted four types of parameter: SF, VI, TF, and CHM from HyMap and ALS with high spectral resolution and high spatial resolution. Of these, ALS data belong to active remote-sensing data, which can reflect the height information of the tree. Hymap's excellent characteristics enable the spectral information corresponding to each pixel to be well reflected. The optimal combination of variables was CHM + VI + SF that could represent the characteristics of each tree species from multiple dimensions. Thus, we can effectively distinguish between different tree species and obtain better classification results. Based on the results of tree-species classification, this study can provide a reliable reference for single-tree-segmentation results. Then the biomass model can be fitted for different tree species to facilitate the subsequent targeted estimation of the biomass of different tree species.

### 4.2. Tree-Segmentation Accuracy Analysis

The spacing distance threshold (SDT), search radius (SR), and overlap were important parameters that affect the segmentation of individual trees. A relatively appropriate threshold can be selected to segment trees in stands with large tree spacing. But it was difficult to determine an appropriate threshold in denser forests. Too-large parameters will result in under-segmentation, whereas smaller parameters will lead to over-segmentation, according to Li et al. [34].

Wang et al. [48] found that the spacing threshold has a significant impact on individual-tree segmentation results. At the same time, the influence of the search radius on canopy segmentation was limited, so the search radius' value should be smaller than the minimum canopy diameter of the plot. Therefore, we adjusted the parameters according to the actual situation of different types of forest plot. From the validation results (Figure 10), it can be seen that the sample plots with high density had difficulties in visual interpretation, and the crown size cannot be accurately depicted. Hamraz et al. [49] developed a generalized single-tree-segmentation method based on small-footprint LiDAR data, which can be applied to natural deciduous forests with complex vegetation structures. No prior knowledge about the tree structure was required to obtain tree height detection. Hence, if a forest has a complex structure in subsequent research, the generalized segmentation method could be used to segment the trees. This will minimize the impact of overlapping leaves.

Besides SDT and SR, the density and integrity of tree point clouds were also important factors that affect the segmentation results. This problem can be ameliorated to some extent by increasing the laser pulse density or using ALS data from multiple overlapping flight paths [50].

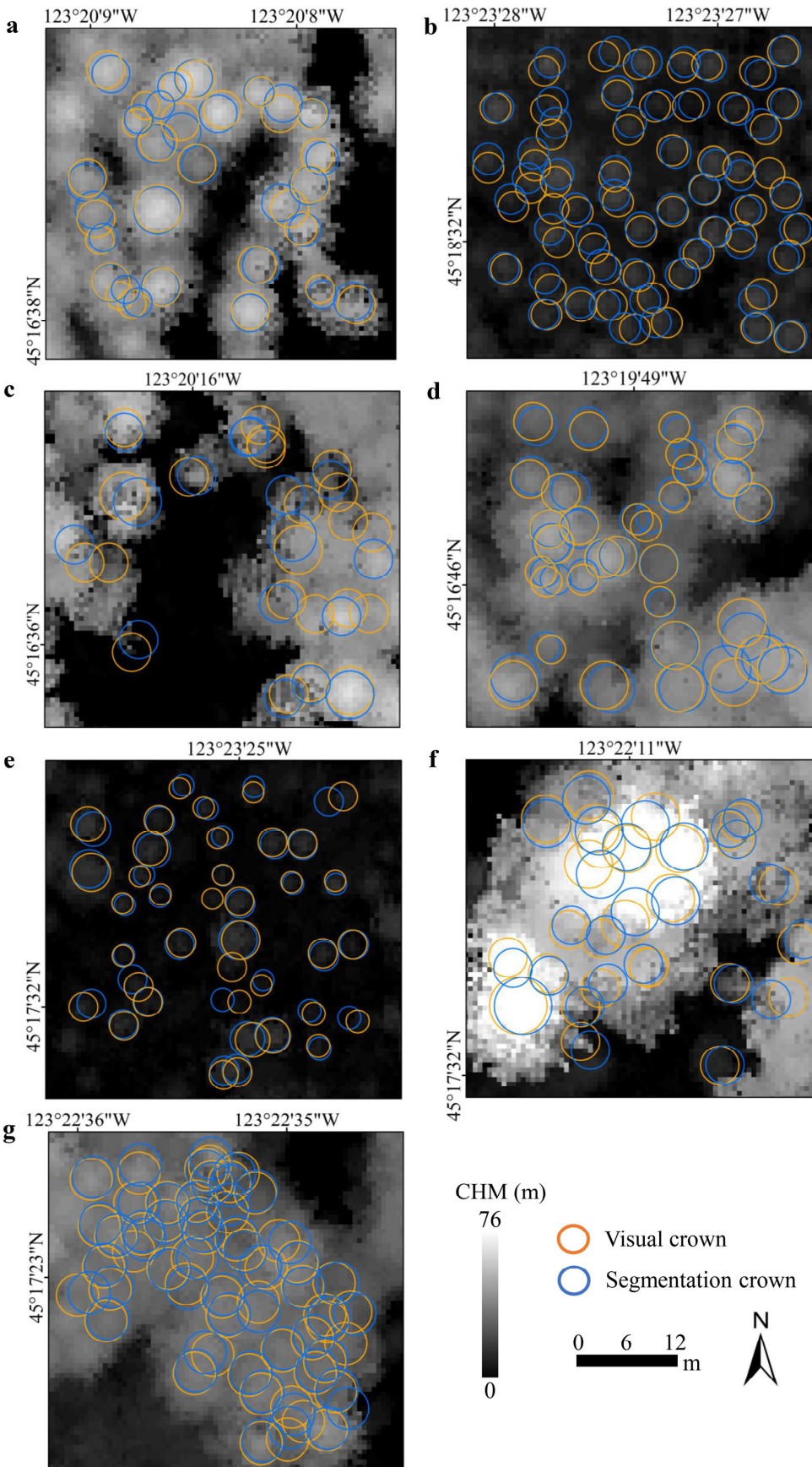

**Figure 10.** The visual-based (i.e., orange circle) and airborne-laser-scanning-based (i.e., blue circle) tree-crown-segmentation results with canopy height model (CHM) in the seven types of forest plot

including PC-MC (**a**), PC-HC (**b**), PC-LM (**c**), PC-MM (**d**), PC-HM (**e**), PC-HB (**f**), PC-LB (**g**). Each tree crown was approximated by a circle. (L: low density; M: medium density; H: high density; C: conifer; B: broadleaf; M: mixed forest.)

*4.3. Effects of Crown Parameters on Biomass Models*

Nelson et al. [51] extracted tree height and canopy factors based on LiDAR data, and then estimated forest biomass and stock volume to obtain better evaluation accuracy. With the advent of commercial ALS [52,53], large-scale forest-resource surveys were convenient and feasible. At present, ALS point cloud data can be used to obtain a high precision of tree-segmentation results, and higher-precision single-wood factors can be extracted from single-wood point cloud results [54]. Since the overall shape of the individual-tree-crown point cloud after tree segmentation was very irregular, it was not possible to calculate it by the approximate regular volume formula. After cutting it into several layers, we found that the shape of each part was similar to a cone or a circular truncated cone. Therefore, the volume of tree crown would be more accurately obtained by step calculation. When crown projection area and crown volume were introduced into the model, the quality of the biomass model can be significantly improved [19]. In this article, it can be seen from Table 6 that the model $R^2$ of Douglas fir, Red alder, and Bigleaf maple were all improved when we added the crown factor to the model.

Douglas fir belongs to coniferous forest, and Red alder and Bigleaf maple belong to broad-leaved forest. According to the results, it could be found that the addition of the crown factor can significantly improve the accuracy of Red alder and Bigleaf maple. For Douglas fir, there was also an increase, but only a small change. This showed that the crown of a broad-leaved tree had a great impact on the biomass of the whole tree. In addition, the biomass-model-fitting result of Douglas fir was excellent, and the RMSE was relatively low, which may be due to its better tree-segmentation results. Due to the large undulating terrain and the large vegetation density in the broad-leaved-forest and mixed-forest plots in PC, the effect of tree segmentation will be worse than that of neatly distributed plantations. Therefore, the precision in parameter extraction was relatively low.

## 5. Conclusions

In this study, we used the canopy-height model, spectral features, and vegetation indices based on the random forest algorithm to classify tree species in PC. According to the results of tree-species classification, the point cloud at the plot scale was further segmented for an individual tree to produce structural parameters. Finally, biomass models of three species were constructed. The importance of this study is to link tree-species classification and biomass model construction, which can build models of different tree species more accurately. It has the potential to be applied to large-scale forest-species management and biomass dynamic monitoring. According to the research results, we concluded the following.

1. The selection of classification variables was critical for tree-species classification. The type and number of variables would affect classification results, and more variables have not produced better results. Variables could be selected based on the realities of the study area and the characteristics of trees.
2. The tree height and crown size extracted by the algorithm were compared with the measured tree height and the results of visual interpretation. We found that they were consistent with the actual situation of trees in PC. The individual-tree parameters measured by ALS in this study could be used as variables in the biomass model.
3. When tree height, crown size, projected area, and volume were introduced as variables into the biomass model, the fitting effects of the three tree species were all optimal. Thus, the introduction of crown parameters into biomass model construction was a feasible method to improve the estimation accuracy of forest individual-tree biomass.

**Author Contributions:** Conceptualization, G.Z., and Y.Q.; methodology and software, Y.Q.; data collection, L.M.M.; data process and analysis, Y.Q.; writing—original draft preparation, Y.Q.; writing—review and editing, G.Z., Z.D., X.M., J.L.; project administration, G.Z.; funding acquisition, G.Z. All authors have read and agreed to the published version of the manuscript.

**Funding:** This research was supported by the National Key R&D Program of China (NO: 2022YFF1303102). This research was funded by the National Science Foundation of China (NSFC) (NSFC award #3201101093 and #42171340), The Precision Forestry Cooperative provided the lidar data at the University of Washington.

**Conflicts of Interest:** The authors declare no conflict of interest.

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
