# Peer review of "Tree-Species Classification and Individual-Tree-Biomass Model Construction Based on Hyperspectral and LiDAR Data"

_remotesensing, doi:10.3390/rs15051341_

Round 1

Reviewer 1 Report

General comments

The paper deals with the classification of three tree species, which were then used for biomass modelling. Four groups of different variables were used for this purpose, so that it was possible to investigate and evaluate the variables useful for such a study. However, the manuscript raises a number of questions to which I would like to know the answers. Among other things, why was species classification not done on individual trees if biomass estimation was already done on individual trees? There is also a lack of information on how the trees were extracted for species classification. Why did the study not use structural and intensity variables directly from the point cloud, but only from the CHM? This could have benefited the species classification and improved accuracy, which is not surprising. There is a lack of emphasis on novelty. What did the authors do differently and additionally compared to other authors? For more specific questions, see the comments below.

Comments referenced to line numbers

The introduction should be more expansive in the topics you address. A separate paragraph could be devoted to tree species classification and a separate one to biomass modeling. Then what other authors have done as you have written from Line 57.

Line 20 – With ALS data, you could achieve much higher accuracy with additional variables. There is a lot of work on integrating ALS data with optical data and on more tree species, where similar or even better accuracy has been achieved.

Lines 31-41 To rephrase. You start about biomass first, to then talk about individual trees and then biomass again. I would start more generally, going one by one to individual trees, and then biomass.

Line 32 - Something wrongs here, do you mean ‘forest resource monitoring’?

Line 48 - Relative to which data? Multispectral and LiDAR data are usually cheaper.

Line 52 – More common name is "Airborne laser scanning".

Lines 57-59 – In that study, they used many more variables than just the normalized vegetation index. Please elaborate.

Line 70 – What data did they use?

Line 89 – Change „research” to „investigate”

Line 95 - Study area

Lines 101-104 - A reference to this would be useful.

Line 115 - Add that the parameters of the flight are in Table 1

Table 1 – What was overlap for ALS data, and side/forward overlap for hyperspectral data? What was vertical and horizontal accuracy of ALS data?

Lines 123-126 – Please use the passive voice here and throughout the text.

Line 127- Based on which species shares was it determined that this was a mixed forest? 50%/50%? Please specify in more detail.

Table 2 – The underlining in the table is in the wrong place (PC19)

Figure 2 How are you validating structural parameters using field-measured trees?

Line 157 - Why the tree species classification was not done on individual trees if then the biomass estimation was already done? This is very incomprehensible. Moreover, nowhere in the text does it say how these trees were extracted for species classification, how many trees were counted in the sample for each species, etc.

Line 157 – What parameters in Random Forest algorithm have you used? Did you make grid search for best parameters? This information is missing and brief description of RF aswell.

Lines 159-163 Having hyperspectral data, NDVI705 is a more accurate variable for vegetation identification than basic NDVI. Consider this in the future. By the way, I think this step was unnecessary having Canopy Height Model from ALS data, since we can only use ALS points classified as vegetation to develop CHM.

Lines 167-173 The composition of variables is somewhat limited when it comes to hyperspectral data. Why didn't you decide to count and use MNF transformation? MNFs are successfully used in tree species classification.

Table 3 Having hyperspectral data, the information that a green band, for example, was used does not contribute much. It would be better to add information about what spectral range for a particular band was used (e.g. (B1(405nm)).

Line 185 – All variables were counted in ENVI?

Line 191 – The description of variables from the CHM is missing, and I don't see any description of spectral variables. If I am wrong I just kindly ask you to point out where.

Line 204 – Please always do this very carefully. A removed variable even of low importance (on the verge of rejection) can affect the overall accuracy score by up to 2-3%. It is worth checking this before removing.

Lines 213-214 – How did you optimize these parameters?

Lines 230-233 – Have you tried to develop a universal model, without dividing it into slices? If so, how did it correspond with the current one?

Line 247 – „The CAR model was stable and had good estimation accuracy”. This can be stated about many methods, so any proof of the validity of this statement is lacking.

Table 4 – Why different models for different species in different combinations? There is one model for douglas fir, and three each for the other species, in addition to three different ones (live, dead, in bark, etc.).

Lines 275-276 – In Table 1, HyMap's spatial resolution is given as 2.9m, while here it is 0.5m.

Lines 275-279 – How does this entire paragraph relate to what is further described in this subsection? In addition, you wrote „Figure 4 showed the accuracy results of Douglas fir, Red alder, and Bigleaf maple were classified with only CHM.”. Various combinations of variables are given in the figure. All of this is not consistent with each other.

Figure 4. The significant drop in accuracy with textural variables is puzzling. Adding additional variables should not affect such a significant drop in accuracy. How are you able to explain this?

Line 283 – Figure 6a

Lines 283-288 – As I said before, it is necessary to add the spectral range of the band. Hyperspectral scanners with even better spectral resolution have more bands, so the information that it is the N channel, does not allow you to easily repeat and use your research.

Line 296 – I cannot understand which variables belonged to CHM

Figure 5 What was the correlation between the variables NDVI1 and NDVI2? Between the variables B13 and B14? Did you assume any correlation threshold above which correlated variables were excluded?

Line 307 – Figure 6b?

Lines 311-316 I cannot agree with this by referring to what you have shown in Figure 6a. Alder, in the spectral range of 0.8 to 1.2 μm, is significantly different from the other two species. This also confirms the need for variables from ALS.

Figure 6c How many hyperspectral data scenes were used to map the entire area? You can clearly see the differences between the scenes, which raises concerns about whether the proposed solution can be useful in practice.

Lines 331-336 For what purpose did you make such a detailed division?

Table 5 – Tree detection rate – how was it calculated? There is no information about it in the methodology.

Lines 356-369 Was there a normal distribution for individual tree species, height and crown area? In some cases, it looks like there was a significant accumulation of trees with similar heights, and there were missing samples for trees with varying height or area parameters.

Figure 9 Why are there only 30 samples on the chart, and not all of them for each species? And here again the question is how many of these samples were finally for each species? Missing from the figure is the addition of letters (a,b,c). In the third figure one tree has a height of 5 m as far as field measurement is concerned and from your results more than 15 m, where do such discrepancies come from? Such cases significantly affect the reduction of R2.

Table 6 Douglas fir unnecessarily bold.

Discussion - The discussion in general is well structured, with many references to other authors, but please put emphasis on what is novel in your study.

Lines 415-419 – It is agreed that structural information from ALS may not be sufficient to correctly identify species, but in your study you used only CHM-based variables, without counting structural variables from the entire point cloud.

Lines 438-440 – This sentence is confusing. Try to rephrase it.

Lines 447-450 You suggest that textural variables can be used in species classification with other variables, but when combined with all variables, this group of variables degrades accuracy. I would be more inclined to write that these variables need to be omitted, or explored even more in search of others from this group. The fact that something worsens the result is also important information.

Line 451- CASI. An abbreviation that was previously unexplained

Lines 470-471 – Empirical evidence of this in your study is lacking.

Line 476 – Change can’t to can not.

Lines 507-508 Unintelligible sentence.

Line 516 – Delete „And”

Conlusions - Please elaborate according to the revised text.

References – Note that some journal names or titles are not written according to the publisher's requirements

Reviewer 2 Report

The aim of this paper is to combine active (LIDAR) and passive (hyperspectral) remote sensing data to realize tree species classification and establish tree species biomass models, in a study area in Panther Creek, Oregon of northwest USA. The paper is not innovative, and the structure needs improvements. The state-of-the-art is short but comprehensive and the aims were clear. The methodology is fair, following known methods for LIDAR and optical image processing, and combining them for modeling Trees biomass. I have a main question about the analysis and use of variables in the model. Moreover, the results don’t present the final maps with the quantification of AGB, obtained by the combination of the different data and techniques (LIDAR, Hyperspectral, Modeling). Despite I consider the paper needs major improvements I make some detailed comments:

P 6 - Table 3 – improve the explanation about why did you use these vegetation index and texture features derived for Hyperspectral images

P 7, L 192 – “These features may be highly correlated or redundant” – Please present the correlation matrix of Pearson

P 8, L 250 – for the Individual tree biomass model the authors used the variables: “crown size, crown projected area, and crown volume”. What is the Pearson correlation between these variables? Did you check the collinearity?

The author should provide supplementary material with the statistics of regression model adjustments.

Reviewer 3 Report

1. L82: Currently, most studies .. include citation for this statement, which studies are referred to?

2. Objectives should be written in sentences instead of point forms.

3. Figure 4 should be presented in bar chart rather than a connected points as the variables are not related to each other.

4. Conclusion should also be written in sentences, rearrange the bullets into continuous sentences and keep the continuity.

5. Include significances of the findings, why it is important and who will benefit from the study. 

Round 2

Reviewer 1 Report

The authors have paid a lot of attention and significantly enriched their manuscript with additional information. I am satisfied with the responses provided and accept the article as presented.

Reviewer 2 Report

Thanks you to the authors for the answers and improment of the paper.

I maintain my inicial apreciation. This papper dosnt not bring a significtan novelty, but is fair for publication in a journal.

Best regadrs.